# Rotational Spectrum and Conformational Analysis of Perillartine: Insights into the Structure–Sweetness Relationship

**DOI:** 10.3390/molecules27061924

**Published:** 2022-03-16

**Authors:** Gabriela Juárez, Miguel Sanz-Novo, José L. Alonso, Elena R. Alonso, Iker León

**Affiliations:** Grupo de Espectrocopía Molecular (GEM), Edificio Quifima, Laboratorios de Espectroscopia y Bioespectroscopia, Unidad Asociada CSIC, Parque Científico UVa, Universidad de Valladolid, 47011 Valladolid, Spain; gabriela.juarez@uva.es (G.J.); miguel.sanz.novo@uva.es (M.S.-N.); jlalonso@qf.uva.es (J.L.A.); elenarita.alonso@uva.es (E.R.A.)

**Keywords:** perillartine, high-intensity sweetener, Fourier transform microwave spectroscopy, Shallenberger–Acree–Kier theory

## Abstract

We used high-resolution rotational spectroscopy coupled to a laser ablation source to study the conformational panorama of perillartine, a solid synthetic sweetener. Four conformers were identified under the isolation conditions of the supersonic expansion, showing that all of them present an *E* configuration of the C=N group with respect to the double bond of the ring. The observed structures were verified against Shallenberger–Acree–Kier’s sweetness theory to shed light on the structure–sweetness relationship for this particular oxime, highlighting a deluge of possibilities to bind the receptor.

## 1. Introduction

The sense of taste is one of the most important classic senses developed during evolution, which first appeared in the microbial world two billion years ago [1,2]. Humans can distinguish at least five basic taste qualities: sweet, bitter, sour, umami, and salty. Among them, the sweet taste sensation is innately pleasant as it is, evolutionarily speaking, correlated with the presence of nutritious foods. Several studies suggest that humans predicate this flavor from birth [3,4]. This attraction for sweetness is so strong that the abundance of sugar in the modern diet has contributed to the obesity and various diseases associated with this condition, which is the reason why some specialists suggest promoting healthier options, such as the consumption of sweeteners [5,6,7]. A sweetener is a food additive that provides a sweet taste similar to sugar. It can be classified according to its origin as natural or artificial sweeteners. For instance, carbohydrates are the most important family of the former, whereas saccharin could represent the latter [8,9].

Compounds with unrelated structures have shown sweetness; it has been difficult to accommodate them within a unified theory. The sensation of sweetness begins on the tongue and soft palate, where the taste receptors, T1R2-T1R3, present specific binding sites that interact with sweet compounds [10,11,12,13] that must have characteristic binding points [10,11,12,13,14]. To rationalize the sweet taste, Shallenberger and Acree proposed a simple theory in which sweet substances have specific sites to interact with the receptor system [15]. A sweet molecule presents an AH/B system that establishes two hydrogen bonds through a complementary AH/B system at the receptor located in the taste buds. A and B are electronegative atoms (O, N, Cl, S among others). The A-H is, thus, a group that can form hydrogen bonds, while B is a good proton acceptor atom. This AH/B pair is known as “glucophore” and is the heart of this theory. The distances between the AH/B atoms range from 2.5 to 4 Å [16,17]. A refinement of this theory, proposed later by Kier [18], introduces a third interacting site known as γ. This new point must be hydrophobic or hydrophilic to interact with the receptor via dispersive forces. This three-point contact AH/B/γ model is known as ‘the sweetness triangle’, and it was crucial to explain the stereo-selectivity of the receptor [18] (see Figure 1).

In this regard, deciphering the shape of the target molecule is essential to understanding its interactions with the receptor. Unfortunately, such studies are very challenging due to the distorted geometries in the condensed phase. Furthermore, the surrounding media usually perturbs the inter- and intramolecular interactions. An alternative is to study the target molecule in the gas phase, more precisely, under supersonic expansions, where an isolated environment is created. In order to carry out such a task, another problem must be overcome, i.e., sweeteners usually present a high melting point and cannot be heated without decomposing; conventional heating methods for vaporization cannot be employed. Fortunately, we can use high-energy laser pulses in a controlled way to solve the problem of driving thermolabile species into the gas phase. In recent years, we combined laser ablation and advanced Fourier transform microwave (LA-CP-FTMW) techniques in supersonic expansions to conduct several gas-phase studies of sweeteners. The high resolution and robustness of this technique allowed us to identify their conformations with very accurate 3D structural information, as demonstrated in previous studies on sugars [19,20,21,22]. Altogether, it is possible to shed some light on the long-considered question of why these molecules are perceived as sweet [13,15,23,24]. The first sweeteners analyzed were natural sweeteners, such as fructose or tagatose [25,26]. Subsequently, the study was extended to different polyols, such as dulcitol and sorbitol [27], and the research was continued with one of the oldest synthetic sweeteners, namely, saccharin [28].

To further rationalize the theory of sweetness and extend it to sweeteners involving functional groups different from those of the molecules mentioned above, we have investigated perillaldehyde oxime (perillartine). This synthetic sweetener is especially attractive given its intensely sweet taste, approximately 2000 times sweeter than sucrose [29]. This peculiar molecule, which is easily obtained by the oximation of perillyl aldehyde, is an oxime-containing cyclohexene where the double bond is conjugated with the C=N double bond, as shown in Figure 2. Perillartine is used commercially in Japan to sweeten tobacco, but it lacks further applications due to its low solubility and a lingering ‘metallic’ taste [30]. It is worth noticing that this compound exhibits several possibilities for the AH and B units [31]. Shallenberger & Acree initially suggested the hydroxyl moiety as AH and the delocalized cloud of the ring as B (see Figure 2a). Later on, Kier selected one of the hydrogen atoms in the *ortho* position with respect to the oxime group as AH and the hydroxyl oxygen as B, as shown in Figure 2b, which is in good accordance with several studies of oxime derivatives of perillartine [18,32,33]. Alternatively, Heijden et al. [34] proposed that both AH and B identities are located within the oxime (Figure 2c). Although some studies have been carried out on perillartine and its analogs [32,35,36,37,38,39], such as the analysis of electrostatic potentials [38], the three-dimensional structure has never been specified.

In this context, we have performed the first rotational spectroscopic study of perillartine using LA-CP-FTMW spectroscopy to identify the most relevant conformers of this artificial sweetener. These precise structural results will be compared with the proposal of Shallenberger–Acree–Kier’s sweetness theory.

## 2. Results and Discussion

### 2.1. Conformational Space

We explored the conformational space of perillartine using molecular mechanics methods. For the conformational search, the Merck Molecular force field (MMFFs) and two search algorithms, the “large scales low mode” (which uses frequency modes to create new structures) and a Monte Carlo-based search, as implemented in Macromodel [40], were used. A total of seventeen structures of perillartine were obtained. These geometries were submitted to Gaussian suite programs [41] for subsequent geometry optimizations. The models of choice were the Møller–Plesset (MP2) perturbation theory in the frozen core approximation [42] and the B3LYP density functional, including the Grimme dispersion [43,44] with Becke–Johnson damping [45,46], both with the Pople’s 6-311++G (d,p) basis set. [47] Frequency calculations were also computed to ensure that the optimized geometries were true minima, as well as to calculate the zero-point and Gibbs free energies. After this process, a total of 12 different structures were obtained (see Figure 1 and Appendix A). The most relevant spectroscopic parameters, such as the energetics, rotational constants, and electric dipole moment components using B3LYP-GD3BJ/6-311++G(d,p) are listed in Table 1. The values obtained using MP2/6-311++G(d,p), as well as a small discussion of the different methodologies, can be found in the Appendix A.

The twelve conformers of perillartine can be categorized into different groups depending on the position of the allyl and oxime groups. We have used a combination of three symbols to label these conformers: the configuration between the C=N group and the double bond in the ring can be oriented in a *trans* or *cis* position and is indicated using the capital letters *E (entgegen)* or *Z (zusammen),* respectively. A lowercase letter *e* (*equatorial*) or *a* (*axial*) indicates the orientation of the allyl group. Finally, the Roman number designates the energy order for the different conformations associated with the different dispositions of the allyl group.

The six most stable structures have an *E* configuration between the C=N group and the double bond in the ring, with the oxime group in an *anti* disposition. As expected, the *Z* arrangement is greatly destabilized compared to the *E* disposition, and the most stable conformer with a *Z* arrangement is predicted ~1200 cm^−1^ relative to the global minimum. Accordingly, only four structures are below 200 cm^−1^ and are expected to be observed in a supersonic expansion. This conformational panorama is in good agreement with that observed for perillaldehyde [48].

Additionally, we also estimated the barrier for the internal rotation of the methyl group. The obtained barrier height is ~550 cm^−1^ and, therefore, no rotational splitting is expected (see Appendix A).

### 2.2. Rotational Spectrum and Conformational Identification

The broadband microwave spectrum of perillartine between 2.4 and 8.0 GHz is shown in Figure 2. We first removed the spectral signatures of known photofragmentation products [49,50,51] and turned our attention to the calculated spectroscopic parameters of the most energetically-stable conformers. These structures are nearly prolate asymmetric rotors exhibiting a sizable *µ_a_* electric dipole moment. It eased our analysis since characteristic bunches of *µ_a_*-type *R*-branch lines at a regular *B* + *C* spacing should be easily recognizable in the broadband spectrum. Three progressions were soon identified as belonging to three different rotameric species labeled I, II and III (see Figure 2). Note that *µ_b_*-type *R*-branch lines were only observed for rotamer II. The individual lines appeared broadened by the effect of the ^14^N nuclear quadrupole coupling interaction, not resolved by this broadband technique. The measured transitions were fitted using a rigid rotor Hamiltonian [52]. The resulting rotational constants are listed in the first three columns of Table 2.

Afterward, we removed the rotational lines from the above-assigned rotamers, and, despite its low intensity, we recognized a fourth weaker *µ_a_*-type *R*-branch progression belonging to a fourth rotameric species, labeled as IV (see Figure 2). Subsequently, we looked for *b-* or *c*- type transitions without success. The rotational constants of rotamer IV obtained after the fit are listed in the fourth column of Table 2. Finally, after removing the transitions of this new rotamer, no spectral features attributable to other conformers of perillartine remained in the spectrum. The complete list of measured transitions for the four identified species is collected in Appendix A.

The assignment of the rotamers to the corresponding calculated conformers relied on a comparison between the predicted and experimental rotational constants. It can be easily seen in Table 2 that rotamer IV can be identified unequivocally as *a-E*-I, since the *axial* configuration implies a significant change in the values of the rotational constants. Afterward, the identification of rotamers I, II and III as conformers *e-E*-I, *e-E*-II, and *e-E*-III was simple thanks to the excellent agreement between the theoretical and experimental spectroscopic parameters, mainly based on the close resemblance of the *B* and *C* rotational constants. Thus, we obtained scale factors by dividing the experimental rotational constants by the calculated ones; the scale factors ranged from 0.971 to 1.020, which brings the Density-functional theory (DFT) predicted values of the rotational constants into coincidence with the experimental ones. This remarkable theoretical-experimental synergy has been found in previous rotational studies [27,53] and is in the context of a recent spectroscopic benchmark database [54]. The observation of all the low-energy species (the four representative conformers) further supports the global consistency of the conformational assignment.

### 2.3. Perillartine and the Theory of Sweetness

Once the conformational landscape of perillartine was characterized and assigned, we then compared the geometries of the detected conformers with those of the tripartite glucophore, following the sweetness theory. As mentioned in the introduction, the identity of the AH, B groups has remained uncertain until now. Nevertheless, some of the plausible selections of the contact points exhibit rather unusual distance parameters when compared to those proposed within the frame of Shallenberger–Acree–Kier’s sweetness theory. Thus, we initially suggested that the hydroxyl oxygen atom corresponds to the proton acceptor point (B), while the ring’s C-H opposed to the C=C is the donor group (A-H), since it is an electropositive site of the molecule. Moreover, this reasoning matches one of the previously proposed AH and B identifications [18], which is in consonance with the experimental observation of the sweet taste in different derivatives of perillartine [32]. It shows that the conjugation of a double bond with the oxime group is a requisite for sweet taste in this particular compound. Figure 3 shows a sketch of the observed structures, and their calculated distances between the AH and B contact points. As can be seen, the distances agree with what is postulated in the sweetness theory. The third contact point (γ) should correspond to the allyl group, since it is a hydrophobic zone. We also note that the γ point is not a point but a region. In fact, this hydrocarbon section is important and can vary drastically, as shown for perillartine analogues [38]. Additionally, the three *equatorial* conformers show a double possibility to bind the receptor due to the planarity of the oxime group conjugated with the double C=C bond of the ring, which supports, even more, the previous statements. As an example, the two plausible sweetness triangles, i.e., the glucophore tripartite, for the most stable conformer are depicted in Figure 4.

We have also estimated the following abundances for the four observed rotamers of perillartine from the relative intensities: *e-E*-I: 32 %, *e-E*-II: 48 %, *e-E*-III: 9 %, *a-E*-I: 11 %. At first glance, these estimations do not match the expected ratios derived from the theoretical computations (see Table 1). This behavior can be ascribed to a possible interconversion between structures *e-E*-III and *e-E*-II (see Figure 5), which may alter their population and which does not enable a direct estimation of their conformational abundances. Fortunately, the conformational interconversion barrier into structure *e-E*-I is significant enough to preclude any interconversion and its population is not affected by other conformers. Thus, our estimations should be focused exclusively on conformers *e-E*-I and *a-E*-I, and set structure *e-E*-I as the global minimum. The *e*-*E*-I is slightly more stable than the other conformers, as it adopts a disposition where the methyl group has a lesser steric effect. Nevertheless, these relaxation issues shall not affect the sweetness of perillartine since, as stated above, all the conformers exhibit a suitable structure in the framework of Shallenberger–Acree–Kier’s theory.

Finally, it is important to note that, to understand why a molecule is sweet, not only does the interaction between the molecule and the receptor need to be taken into account, but also additional factors such as the propensity of the molecule to “reach” (solvent interactions) or “fit” (molecular size) in the receptor. Interestingly, the *axial* and *equatorial* dispositions of perillartine are considerably different, allowing more diversity to fit in the receptor. Additionally, the rich conformational space, in which some of the conformers may be interconverting continuously at room temperature, gives perillartine several shapes, increasing the change to fit in the receptor, as if they were slightly different pieces of a puzzle. Altogether, these properties increase the chance of perillartine to interact with the receptor, making it a particularly sweet molecule.

## 3. Materials and Methods

Solid rods of perillartine (m.p 102 °C) were prepared by using commercial samples without further purification and by pressing the compound’s fine powder mixed with a small amount of a commercial binder. These rods were placed in the ablation nozzle, and a picosecond Nd: YAG laser (355 nm, 16 mJ per pulse, 20 ps pulse width) was used as a vaporization tool. The resulting products of the laser ablation were supersonically expanded by utilizing a flow of neon gas at a backing pressure of 10 bar and then probed by chirp pulsed Fourier transform microwave (CP-FTMW) spectroscopy. Details of the experimental setup have been given elsewhere [55]. Chirped pulses of 4 µs directly generated by the 24 GS s^−1^ arbitrary waveform generator were amplified to about 300 W peak power using a traveling wave tube amplifier. The resulting pulses were then transmitted and detected by broadband microwave horn antennas in a high-vacuum chamber, where they interacted with the molecular supersonic expansion. At a repetition rate of 2 Hz, 10.000 free induction decays (4 FID emissions per gas pulse), each with a 10 µs length, were averaged and digitized using a 25 GS s^−1^digital oscilloscope. The frequency-domain spectrum in the 2–8 GHz frequency range was obtained by taking a fast Fourier transform (FFT) following a Kaiser–Bessel window to improve the baseline resolution. The estimated accuracy of the frequency measurements is 30 kHz.

## 4. Conclusions

The molecular shape of perillartine was investigated in the gas phase using rotational spectroscopy. Four distinct structures were identified in the supersonic jet, showing an *E* configuration of the C=N group with respect to the double bond in the ring. Among them, one of the structures has the allyl group in an axial position, while three structures show the equatorial disposition with different orientations of the allyl group. Nevertheless, all the experimental conformers seem to have a three-point contact, as proposed by the sweetness theory. Moreover, despite the assignment of the AH-B moiety of the molecule has remained unknown so far, the precise determination of the three-dimensional structure derived from the rotational constants further points to the ortho H of the ring as the AH entity and the hydroxyl oxygen atom as the B point, respectively. Additionally, the rich conformational panorama highlights the existence of a plethora of possibilities to bind to the receptor and to help us further rationalize the intensely sweet taste of this particular oxime. In general, our work provides additional and valuable support to Shallenberger–Acree–Kier’s theory that links the chemical structure of conformers and sweetness.

## Data Availability

Not applicable.

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
