# Peer review of "Rotational Spectrum and Conformational Analysis of Perillartine: Insights into the Structure–Sweetness Relationship"

_molecules, 2022, doi:10.3390/molecules27061924_

Round 1

Reviewer 1 Report

This manuscript describes the gas phase conformational space of perillartine, a synthetic sweetener, using broadband rotational spectroscopy aided by quantum mechanical calculations. Transitions belonging to the four lowest energy conformers were identified in the rotational spectrum and based on the geometry of the identified species, the authors carried out an analysis of the Shallenberger-Acree-Kier’s theorem to shed light on the relationship between the molecular structure and sweetness of perillartine. The most significant finding of the work is that the complex conformational landscape of perillartine increases it chances to bind to the taste receptors making it a particularly sweet molecule. The work is well-written and concise, and I believe the data will be of interest to the readers of Molecules. Therefore, I recommend the paper for publication.

I do have some comments/suggestions for the authors to consider:

Introduction:

  1. Line 41: “B is a good electron acceptor atom” - Please double check whether “electron” is correct in here, or if it should be “proton” instead.
  2. Line 43: The word “range” appears twice. I feel like the second one is not necessary.
  3. Line 76: “Scheme 1” should be replaced by “Scheme 2”
  4. Line 89: Scheme 2 - caption – The given caption does not match the figure. It seems like it describes Scheme 1 again. Please correct the caption of Scheme 2.

Results and Discussion

  1. Line 106: In the SI, the authors mention [B3LYP-D3(BJ)] in the caption of the Cartesian coordinate tables, but do not mention in the main article text. Please include the references to the Becke-Johnson damping function and update the text accordingly to read “B3LYP-GD3(BJ)” instead of only “B3LYP-GD3”.
  2. Table 1: Please indicate whether the values of the dipole moment components are absolute or not. If not, please indicate their respective signs.
  3. Paragraph starting on Line 127: It would be easier for the readers to visualize the cis-trans, axial-equatorial orientations if the figure showing the 12 conformers were moved to the main article. Representing the double bonds in the 3D molecular structures of Figures 2 and S1 could additionally improve this visualization.
  4. Line 182: Please clarify in the text how the scaling factors were obtained.
  5. Line 220: The authors talk about the derived conformational abundances of the identified conformers in the supersonic jet and comment on their differences with their predicted ratios from Table 1. Including the populations (%) of each conformer in Table 1 (Boltzmann distribution at 298K) based on the relative E, E+ZPE and G energies will help the readers to compare better the experimental and theoretical abundances and to understand the conformational cooling and potential population transfer in the supersonic jet.
  6. The authors could add a brief statement saying that no splittings related to methyl internal rotation has been observed.
  7. Why have not the authors included any distortion constants in their fits on Table 2?
  8. Is there any simple explanation of why conformer e-E-I is the most favored geometry? If yes, a brief explanation could be added to the manuscript.

References

  1. Please double check the format of the references. I noticed that some of them are not in the correct format such as reference 34 – Macromodel.

Reviewer 2 Report

Summary
The authors present a detailed study of the high-resolution rotational spectrum perillartine. They report four observed conforms in the gas-phase study. These are compared to DFT calculations. The observed structures are then further compared to the binding receptors in humans that would detect sweetness in molecules.

In general, the authors find that the most stable conformers observed have structures that would make them ideal for binding in the sweeteners. In general, I find it to be a well written and informative paper that connects gas-phase spectroscopy to the chemical properties of the receptors of taste. I have a few comments below that deal mostly with the benchmark values present or not presented. Otherwise, I find the manuscript suitable for publication.
Comments
There is a previous study on the rotational spectroscopy sweeteners, natural sweeteners, in the detection of ribose. Since the authors want to give credit to the first study, they should consider citing the ribose paper.

The accuracy of the instrument used is listed as 10 kHz. The authors report fits with σ values over 20 kHz. They also report using only a rigid rotor fit for 4 observed conformers. Certainly the inclusion of centrifugal distortion constants would improve these fits. Moreover, while the constant show agreement to the DFT predictions, there are also predicted distortion constants to compare to. Even for harmonic approximations, the CD terms are at least correct in order of magnitude and sign, and would help to
confirm the assignments with an additional value.

Continuing with the predictions, it is great to see that there are comparisons done at some point between MP2 and B3LYP. But are the MP2 ever compared in the paper? If not, why are they mentioned? It would be a good benchmark comparison to include. This is actually a complaint I have with many authors, not just this work, and I try to point this out whenever I can. Please try not to bring up other calculations if the data is never presented. Don’t tease me like that.

When using the scale constants, how do these compare to other sugars and calculations? Perhaps it would be worth the time to compare to the scale constants found for a series of molecules, shown in the Oswald and Suhm paper. (Phys. Chem. Chem. Phys., 2019, 21, 18799-18810)

Again on the scale constants, how is the range of values obtained? are they averaged per conformer over the three rotational constants, or average per rotational constant over the four conformers?

Figure 1 calls the assignments by rotamers, but other places they are referred to as conformers. I understand that they mean conformational isomers and roational isomers, but perhaps the language can be unified? Not a big deal if not.

In line 137, the authors note that the Z isomers are at least 500 cm-1 above the E. this is true, but I think they could be more accurate. The lowest energy Z conformer (eq) is at least 500 cm-1 above the next lowest E conformer (ax), but should we compare similar features as much as we can. In this case the next lowest energy E conformers with with eq structure is e-E-III, about 1200 cm-1 lower.

Minor comments
Page 1, paragraph 2, O, N, Cl, S among others.
page 2, line 53 the environment is really virtually isolated? or is it simply, isolated?
page 2, line 55 it might be more clear to say traditional heating methods for vaporization.

Reviewer 3 Report

Comments to the Author

Report on: Rotational Spectrum and Conformational Analysis of Perillartine: Insights into the Structure-Sweetness Relationship

submitted to Molecules by Juárez-López et al.

The paper reports the investigation of the perillartine molecule and its properties as a synthetic sweetener on the basis of the Shallenberger-Acree-Kier’s sweetness theory. The authors describe the theory and the properties that sweet substances have to fulfill in order to interact with the receptor system, explaining clearly the AH, B and γ points. Three possibilities for perillartine are proposed in which the AH and B points are identified to different parts of the molecule. Then, the authors present the theoretical results from the rich conformational space of perillartine and compared with the broadband rotational spectrum. Several characteristic bunches of lines were recognized and later assigned to four of the theoretical structures. Finally, the authors analyze the shape of the most stable rotamer of perillartine to identify the possible AH and B interaction points.

The paper is well written, the results and the discussion are well explained and there is a good selection of the previous bibliography. I have a just few questions and comments, which could improve the paper before its publication.

-  The first sentence of the introduction is indeed a strong statement and should be supported by a reference.

- In the conformational space section, the authors mention that they have performed geometry optimizations for 17 structures using both MP2 and B3LYP. However, in the remaining manuscript and in the SI there is no other mention to the MP2 calculations. Why did the authors give only the results from B3LYP?

- The authors have observed μa-type transitions for the four detected conformers of perillartine, and μb-type only for e-E-II. Based on the theoretical electric dipole moment components from Table 1, it could be expected also some μc-type transitions for a-E-I. They state they have looked for such transitions, without success. Do they have an explanation for this? Do the MP2 also predict a sizable μc electric dipole moment for a-E-I?

- The estimated abundances for the four rotamers of perillartine are given from the relative intensities. Apparently, those estimations do not match with the ratios obtained from theoretical calculations, but the theoretical values are not given. It would be helpful if they include such values.

Typos/style:

- line 37 in the first page “interaction to interact” could be changed to something less redundant.

- line 76 in the second page: In Scheme 1 there is no depict of perillartine. Change to Scheme 2.

- line 144 in page four: the spectrum appears in Figure 1, not in Figure 2.
